# Michael Adduct of Sulfonamide Chalcone Targets Folate Metabolism in Brugia Malayi Parasite

**DOI:** 10.3390/biomedicines11030723

**Published:** 2023-02-27

**Authors:** Priyanka S. Bhoj, Sandeep P. Bahekar, Shambhavi Chowdhary, Namdev S. Togre, Nitin P. Amdare, Lingaraj Jena, Kalyan Goswami, Hemant Chandak

**Affiliations:** 1Department of Biochemistry, Mahatma Gandhi Institute of Medical Sciences, Sevagram, Wardha 411002, India; 2Department of Chemistry, G. S. Science, Arts and Commerce College, Sant Gadge Baba Amravati University, Khamgaon 444303, India

**Keywords:** Michael adducts, sulfonamide chalcone, antifilarial, dihydrofolate reductase, folate metabolism

## Abstract

A series of Michael adducts of malononitrile and sulfonamide chalcones were synthesized, characterized, and evaluated for their antifilarial activity. Out of 14 compounds, N-(4-(4,4-dicyano-3-p-tolylbutanoyl)phenyl)benzenesulfonamide showed favorable drug-likeness properties with marked antifilarial effects at micro-molar dosages. Apoptosis in *Brugia malayi* microfilariae was confirmed by EB/AO staining, MTT assay, and cytoplasmic cytochrome c ELISA. Since chalcone and folate synthesis pathways share the same substrate, we hypothesize a structural analogy-based inhibition of folate metabolism by this compound. Molecular docking against a pre-validated BmDHFR protein showed more favorable thermodynamic parameters than a positive control, epicatechin-3-gallate. The compound significantly suppressed the DHFR activity in a parasite extract in vitro. Our hypothesis is also supported by a significant reversal of DHFR inhibition by folate addition, which indicated a plausible mechanism of competitive inhibition. These results demonstrate that targeting filarial folate metabolism through DHFR with consequent apoptosis induction might be rewarding for therapeutic intervention. This study reveals a novel rationale of the structural analogy-based competitive inhibition of DHFR by Michael adducts of sulfonamide chalcones.

## 1. Introduction

Lymphatic filariasis (LF) poses a risk to about 863 million people in 47 countries worldwide, with a huge burden of consequent disability [1]. Failing in eliminating LF by 2020, the WHO has set a new target of achieving LF elimination by 2030 by introducing triple-drug therapy (IDA): ivermectin, diethylcarbamazine (DEC), and albendazole. However, this preventive chemotherapy strategy is facing many challenges, including setbacks due to COVID-19, poor drug acceptability among communities due to a lack of awareness, and lack of training of health workers [2,3,4]. Moreover, the threat of disease re-emergence due to insufficient mosquito vector control programs and possible drug resistance is a matter of concern. The absence of a prophylactic vaccine complicates it further. All these compelling issues make antifilarial drug research a dire necessity, in consonance with the mandate of drug research for filarial diseases set out by WHO (TDR) [5].

The most accepted mechanism of DEC action entails eliciting a host inflammatory response against the filarial parasite. Although DEC targets the folate pathway as well, its direct role in filaricidal action is still obscure [6]. Evidence suggests the apoptotic effect of DEC on the parasite in vitro [7], although it is not sufficient for filaricidal action. In our previous work, certain herbal extracts rich in polyphenolic/flavonoid ingredients have shown efficacy against *Brugia malayi* parasites [8]. Since the shikimate pathway of folate and flavonoid biosynthesis shares a precursor called chorismite [9], a possible structural resemblance-based inhibition of folate metabolism by flavonoids through dihydrofolate reductase (DHFR) inhibition can be contemplated. A similar rationale with consequent apoptosis was demonstrated with the flavonoids derived from tea [10].

Chalcones are flavonoid metabolites that are being extensively studied for their spectrum of actions on various cellular metabolic processes [11]. Interestingly, chalcones and their derivatives have proven antiparasitic effects [12]. Recently, we have shown the antifilarial activity of sulfonamide chalcones with consequent apoptosis induction; however, the mechanism is unclear [13]. Comprehensibly, a close association between folate pathway inhibition and apoptosis can be envisioned due to the crucial role of folate in the nucleic acid synthesis required for cell proliferation. Against this backdrop, we have undertaken this study to ascertain the antifolate effects of Michael adducts of sulfonamide chalcones in the human lymphatic filarial parasite.

## 2. Materials and Methods

### 2.1. Chemistry

All solvents and chemicals were obtained commercially and were used as received. Melting points were determined in an open capillary and were not corrected. The reaction’s progress was checked using pre-coated TLC plates. The synthesized compounds were characterized based on chemical properties and spectral analysis. IR spectra were recorded using a spectrometer instrument (Bruker India Scientific Pvt Ltd., Mumbai, India). NMR spectra were recorded with a Bruker Avance II at 400 MHz as well as a Bruker DMX spectrometer at 500 MHz (^1^H) and 125 MHz (^13^C) using CDCl_3_ or DMSO-*d*_6_ as the solvent (Appendix A) (Bruker Scientific LLC, MA, USA). All chemical shifts are reported in ppm and have been referenced to tetramethylsilane using residual ^1^H or ^13^C signals of the deuterated solvents as internal standards. Electron spray ionization mass spectra were recorded on a Bruker microTOFQ spectrometer (Bruker Scientific LLC, Marlton, MA, USA). Elemental analyses (C, H, N) were obtained using a Carlo Erba 1108 analyzer (EA Consumables, LLC, Marlton, NJ, USA). Synthesis of sulfonamide chalcones (1a-n) was achieved by the Claisen–Schimdt condensation of sulphonamide ketones with substituted aromatic aldehydes [13].

#### 2.1.1. General Method of Synthesis

A suspension of sulfonamide chalcone (1a-n) (1.25 mmol) and malononitrile (1.25 mmol) in the presence of piperidine (0.375 mmol) in 0.5 mL aqueous ethanol (1:1) was stirred at room temperature for an appropriate time. After completion of the reaction, as indicated by TLC, the product was simply separated by filtration under suction to afford a product of sufficient purity. The crude compound was recrystallized in ethanol, if necessary.

##### (4-(4,4-dicyano-3-phenylbutanoyl)phenyl)benzenesulfonamide (**3a**)

White solid; (yield 0.50 g, 94%); mp: 179–180 °C; Rf 0.5 (30% EtOAc:Hexane); IR: 2966 (Ar C-H str), 1593 (C=O str), 1464 (asymm. S=O str), 1156 (symm. S=O str) cm^−1^; ^1^H NMR (400 MHz, DMSO-*d_6_*) (Appendix A). δ: 10.88 (s,1H, NH), 7.86–7.81 (m, 4H, Ar-H), 7.64–7.59 (m, 3H, Ar-H), 7.43–7.41 (m, 2H, Ar-H), 7.36–7.28 (m, 3H, Ar-H), 7.21–7.19 (m, 2H, Ar-H), 5.18 (d, 1H, J = 6.0 Hz, H_1_), 4.00 (m, 1H, H_2_), 3.71 (dd, 1H, J = 21.8 Hz, 9.6 Hz, H_3_) 3.5 (dd, 1H, J = 18.2 Hz, 12.4 Hz, H_4_) ^13^C NMR (100 MHz, DMSO-*d_6_*) (Appendix A) δ: 194.53, 142.35, 139.00, 137.46, 132.98, 130.78, 129.48, 129.16, 128.28, 127.88, 126.36, 117.58, 113.13, 112.84, 48.44, 39.99, 28.88; HRMS (ESI) (Appendix A): *m*/*z*: calculated for C_24_H_19_N_3_NaO_3_S is 452.1045 found 452.1099 [M+Na]^+^.

##### N-(4-(4,4-dicyano-3-(4-methoxyphenyl)butanoyl)phenyl)benzenesulfonamide (**3b**)

White solid; (yield 0.46 g, 81%); mp: 191–193 °C; Rf 0.45 (30% EtOAc:Hexane); IR: 2966 (Ar C-H str), 1590 (C=O str), 1464 (asymm. S=O str) 1228 (symm S=O str) cm^−1^; ^1^H NMR (500 MHz, DMSO-*d_6_*) (Appendix A) δ: 7.87–7.83 (m, 4H, Ar-H), 7.64–7.61 (m, 1H, Ar-H), 7.58–7.55 (m, 2H, Ar-H), 7.35–7.33 (m, 2H, Ar-H), 7.21–7.19 (m, 2H, Ar-H), 6.91–6.89 (m, 2H, Ar-H), 5.13 (d, 1H, J = 6.0 Hz, H_1_), 4.00 (dd, 1H, J = 12.92 Hz, 7.55 Hz, H_2_), 3.94 (dd, 1H, J = 17.05 Hz, 7.9 Hz, H_3_), 3.67 (dd, 1H, J = 17.8 Hz, 11.95 Hz, H4); ^13^C NMR (125 MHz, DMSO-*d_6_*) (Appendix A) δ: 194.40, 159.00, 143.39, 142.78, 136.49, 130.83, 129.55, 129.43, 129.29, 129.15, 126.64, 117.66, 113.79, 113.14, 112.96, 54.80, 40.16, 38.84, 29.21; HRMS (ESI) (Appendix A): *m*/*z*: calculated for C_25_H_21_N_3_NaO_4_S is 459.1253, found 459.3528 [M+Na]^+^.

##### N-(4-(3-(4-chlorophenyl)-4,4-dicyanobutanoyl)phenyl)benzenesulfonamide (**3c**)

White solid; (yield 0.50 g, 82%); mp: 182–183 °C; Rf 0.36 (30% EtOAc:Hexane); IR: 3333 (N-H str), 2879 (Ar C-H str), 1674 (C=O str), 1328 (asymm. S=O str), 1158 (symm. S=O str) cm^−1^; ^1^H NMR (500 MHz, DMSO-*d_6_*) (Appendix A) δ: 10.97 (s,1H, NH), 7.85–7.56 (m, 7H, Ar-H), 7.45–7.20 (m, 6H, Ar-H), 5.17 (d,1H, J = 5.36 Hz, H_1_), 4.02 (m, 1H, H_2_), 3.70–3.76 (m, 2H, H_3_, H_4_) ^13^C NMR (125 MHz, DMSO-*d_6_*) (Appendix A) δ: 194.85, 142.75, 139.28, 136.91, 133.50, 133.06, 130.27, 129.94, 129.63, 128.73, 126.82, 117.96, 113.43, 113.13, 46.11, 39.29, 29.18; HRMS (ESI) (Appendix A): *m*/*z*: calculated for C_24_H_18_ClN_3_NaO_3_S is 486.0655 found 486.0478 [M+Na]^+^ and 488.0465 [M+Na+2]^+^.

##### N-(4-(3-(4-bromophenyl)-4,4-dicyanobutanoyl)phenyl)benzenesulfonamide (**3d**)

White solid; (yield 0.51 g, 82%); mp: 23–212 °C; Rf 0.55 (30% EtOAc:Hexane); IR: 3335 (N-H str), 2963 (Ar C-H str), 1603 (C=O str), 1329 (asymm. S=O str), 1158 (symm. S=O str) cm^−1^; ^1^H NMR (400 MHz, DMSO-*d_6_*) (Appendix A) δ: 7.86–7.83 (m, 4H, Ar-H), 7.66–7.61 (m, 1H, Ar-H), 7.58–7.56 (m, 3H, Ar-H), 7.41–7.39 (m, 2H, Ar-H), 7.22–7.20 (m, 3H, Ar-H), 5.20 (d, 1H, J = 6.0 Hz, H_1_), 4.00 (d, J = 0.92 Hz, 1H, H_2_), 3.73 (dd, 1H, J = 17.95, 9.45 Hz, H_3_), 3.49 (dd, 1H, J = 17.95, 12.25 Hz, H_4_); ^13^C NMR (100 MHz, DMSO-*d_6_*) (Appendix A) δ: 193.98, 142.47, 132.93, 129.55, 128.84, 128.12, 126,38, 118.45, 45.95, 39.85, 26.08; HRMS (ESI) (Appendix A): *m*/*z*: calculated for C_24_H_18_BrN_3_NaO_3_S is 530.0150 found 531.0126 [M+Na]^+^ and 532.0155 [M+Na+2]^+^.

##### N-(4-(4,4-dicyano-3-(3,4,5-trimethoxyphenyl)butanoyl)phenyl)benzene-sulfonamide (**3e**)

White solid; (yield 0.62 g, 96%); mp: 187–188 °C; Rf 0.55 (30% EtOAc:Hexane); IR: 3188 (N-H str), 2966 (Ar C-H str), 1588, (C=O str), 1329, (asymm. S=O str), 1125 (symm. S=O str) cm^−1^; ^1^H NMR (400 MHz, DMSO-*d_6_*) (Appendix A) δ:7.74–7.72 (m, 1H, Ar-H), 7.55–7.36 (m, 5H, Ar-H), 7.26–7.24 (m, 1H, Ar-H), 6.92 (d, J = 8.56 Hz, 1H, Ar-H), 6.82 (s, 1H, Ar-H), 6.66 (d, J = 8.72 Hz, 1H, Ar-H), 6.57 (s, 1H, Ar-H), 5.60 (s, 1H, H_1_), 4.49 (d, J = 11.76 Hz, 1H, H_2_), 4.23 (d, J = 11.76 Hz 1H, H_3_), 4.3 (dd, J = 12.62 Hz, 3.28 Hz, 1H, H_4_), 3.82 (s, 6H, 2(MeO)), 3.70 (s, 3H, MeO); ^13^C NMR (100 MHz, DMSO-*d_6_*) (Appendix A) δ: 197.33, 152.64, 140.52, 139.48, 137.66, 136.61, 132.43, 130.27, 129.76, 128.16, 126.23, 125.79, 120.16, 114.36, 113.80, 36.19, 59.93, 55.84, 55.44, 44.31, 38.95, 22.92; CHN Analysis: Anal. calcd for C_27_H_25_BrN_3_O_6_S (519.14): C, 62.42; H, 4.85; N, 8.09%. Found: C, 62.39; H, 4.83; N, 8.08.

##### N-(4-(3-(2-chlorophenyl)-4,4-dicyanobutanoyl)phenyl)benzenesulfonamide (**3f**)

White solid; (yield 0.43 g, 75%); mp: 181–182 °C; Rf 0.55 (30% EtOAc:Hexane); IR: 3270 (N-H str), 2899 (Ar C-H str), 1663 (C=O str), 1358 (asymm. S=O str), 1156 (symm. S=O str) cm^−1^; ^1^H NMR (400 MHz, DMSO-*d_6_*) (Appendix A) δ: 3.82 (s, 1H, NH), 7.85–7.83 (m, 4H, Ar-H), 7.61–7.57 (m, 2H, Ar-H), 7.54–7.51 (m, 2H, Ar-H), 7.49–7.44 (m, 1H, Ar-H) 7.35–7.28 (m, 2H, Ar-H), 7.24–7.22 (m, 2H, Ar-H), 5.21 (brs,1H, H_1_), 4.54 (d, 1H, J = 6.00 H_2_), 3.75 (dd, 1H, J = 18.06, 8.36 Hz, H_4_), 3.62 (dd, 1H, J = 18.04, 5.06 Hz, H_3_); ^13^C NMR (100 MHz, DMSO-*d_6_*) (Appendix A) δ: 194.06, 142.73, 139.36, 135.15, 133.97, 132.87, 130.73, 129.63, 129.48, 129.44, 129.06, 128.08, 127.41, 126.58, 117.77, 112.51, 40.22, 36.11, 27.75; CHN Analysis: Anal. calcd for C_24_H_18_ClN_3_O_3_S (493.93): C, 62.13; H, 3.91; 9.06%. Found: C, 62.02; H, 3.84; N, 8.96%.

##### N-(4-(4,4-dicyano-3-p-tolylbutanoyl)phenyl)benzenesulfonamide (**3g**)

White solid; (yield 0.38 g, 70%); mp: 191–192 °C; Rf 0.4 (30% EtOAc:Hexane); IR: 3333 (N-H str), 2881 (Ar C-H str), 1677 (C=O str), 1399 (asymm. S=O str), 1157 (symm. S=O str) cm^−1^; ^1^H NMR(400 MHz, DMSO-*d_6_*) (Appendix A) δ: 3.85 (s, 1H, NH), 7.85–7.83 (m, 4H, Ar-H), 7.61–7.51 (m, 3H, Ar-H), 7.32–7.30 (m, 2H, Ar-H), 7.24–7.22 (m, 2H, Ar-H) 7.16–7.14 (m, 2H, Ar-H), 5.12 (d, 1H, J = 5.64 Hz, H_1_), 3.94 (d, 1H, J = 3.88 Hz H_2_), 3.66 (dd, 1H, J = 18.04 Hz, 8.00 Hz, H_4_), 3.47 (dd, 1H, J = 17.88 Hz, 6.00 Hz, H_3_), 2.29 (s, 3H, Me); ^13^C NMR (30 MHz, DMSO-*d_6_*) (Appendix A) δ: 194.48, 142.67, 139.37, 137.42, 134.46, 132.89, 129.94, 129.46, 129.09, 129.04, 127.87, 126.58, 117.79, 113.12, 112.82, 40.21, 38.96, 29.04, 20.63; CHN Analysis: Anal. calcd for C_25_H_21_N_3_O_3_S (443.52): C, 67.70; H, 4.77; N, 9.47%. Found: C, 67.54; H, 4.66; N, 9.44%.

##### N-(4-(4,4-dicyano-3-(4-methoxyphenyl)butanoyl)phenyl)4-methylbenzene-sulfonamide (**3h**)

White solid; (yield 0.49 g, 84%); mp: 162–164 °C; Rf 0.57 (30% EtOAc:Hexane); IR: 3458 (N-H str), 2966 (Ar C-H str), 1671 (C=O str), 1305 (asymm. S=O str), 1151 (symm. S=O str) cm^−1^; ^1^H NMR(400 MHz, DMSO-*d_6_*) (Appendix A) δ: 3.74 (s, 1H, NH), 7.84–7.80 (m, 2H, Ar-H), 7.77–7.71 (m, 2H, Ar-H), 7.40–7.30 (m, 4H, Ar-H), 7.23–7.18 (m, 2H, Ar-H) 6.89–6.83 (m, 2H, Ar-H), 5.3 (d, 1H, J = 5.84 Hz, H_1_), 3.93 (dd, 1H, J = 12.74 Hz, 6.96 Hz, H_2_), 3.75 (s, 3H, MeO), 3.70–3.62 (m, 1H, H_4_), 3.46 (dd, 1H, J = 17.8 Hz, 11.8 Hz, H_3_), 2.34 (s, 3H, Me); ^13^C NMR (100 MHz, DMSO-*d_6_*) (Appendix A) δ: 194.49, 159, 143.39, 142.78, 136.49, 130.83, 129.55, 129.48, 129.29, 129.15, 126.64, 117.66, 113.79, 113.14, 112.84, 54.87, 40.21, 38.96, 29.17, 20.97; CHN Analysis: Anal. calcd for C_26_H_23_N_3_O_4_S (473.54): C, 65.95; H, 4.90; N, 8.87%. Found: C, 65.85; H, 4.86; N, 8.87%.

##### N-(4-(3-(4-chlorophenyl)-4,4-dicyanobutanoyl)phenyl)-4-methylbenzene-sulfonamide (**3i**)

White solid; (yield 0.51 g, 86%); mp: 178–179 °C; Rf 0.67 (30% EtOAc:Hexane); IR: 3443, (N-H str), 2968 (Ar C-H str), 1678 (C=O str), 1333 (asymm. S=O str), 1160, (symm. S=O str) cm^−1^; ^1^H NMR(400 MHz, DMSO-*d_6_*) (Appendix A) δ: 3.72 (s, 1H, NH), 7.83–7.81 (m, 2H, Ar-H), 7.73–7.71 (m, 2H, Ar-H), 7.46–7.44 (m, 2H, Ar-H), 7.37–7.36 (m, 2H, Ar-H) 7.31–7.29 (m, 2H, Ar-H), 7.23–7.21 (m, 2H, Ar-H), 5.15 (d, 1H, J = 6.08 Hz, H_1_), 4.01 (d, 1H, J = 6.56 Hz, H_2_), 3.68 (dd, 1H, J = 17.96 Hz, 9.96 H_4_), 3.51 (dd, 1H, J = 17.96 Hz, 12.36 Hz, H_3_), 2.35 (s, 3H, Me); ^13^C NMR (100 MHz, DMSO-*d_6_*) (Appendix A) δ: 194.15, 143.34, 142.85, 136.46, 136.28, 133.22, 130.66, 129.76, 129.47, 129.38, 128.43, 126.62, 117.62, 112.79, 112.52, 40.22, 38.97, 28.74, 20.98; CHN Analysis: Anal. calcd for C_25_H_20_ClN_3_O_3_S (477.96): C, 62.82; H, 4.22; N, 8.79%. Found: C, 62.64; H, 4.39; N, 8.57%.

##### N-(4-(3-(4-bromophenyl)-4,4-dicyanobutanoyl)phenyl)4-methylbenzene-sulfonamide (**3j**)

White solid; (yield 0.57 g, 88%); mp: 192–194 °C; Rf 0.5 (30% EtOAc:Hexane); IR: 3450(N-H str), 2966 (Ar C-H str), 1679 (C=O str), 1295, (asymm. S=O str), 1161 (symm. S=O str) cm^−1^; ^1^H NMR(400 MHz, DMSO-*d_6_*) (Appendix A) δ: 3.71 (s, 1H, NH), 7.83–7.81 (m, 2H, Ar-H), 7.73–7.71 (m, 2H, Ar-H), 7.52–7.50 (m, 2H, Ar-H), 7.40–7.48 (m, 2H, Ar-H) 7.31–7.29 (m, 2H, Ar-H), 7.23–7.21 (m, 2H, Ar-H), 5.15 (d, 1H, J = 6.24 Hz, H_1_), 3.99 (dd, 1H, J = 13.46 Hz, 7.12 Hz, H_2_), 3.67 (dd, 1H, J = 17.98 Hz, 9.96 Hz, H_4_), 3.50 (dd, 1H, J = 18.06 Hz, 12.32 Hz, H_3_), 2.35 (s, 3H); ^13^C NMR (100 MHz, DMSO-*d_6_*) (Appendix A) δ: 194.09, 143.32, 142.85, 136.7, 136.44, 131.36, 130.61, 129.44, 129.35, 126.6, 121.64, 112.73, 112.47, 40.23, 38.98, 28.63, 21.00; CHN Analysis: Anal. calcd for C_25_H_20_BrN_3_O_3_S (522.41): C, 57.48; H,15.30; N, 8.04%. Found: C, 57.21; H, 3.52; N, 7.89%.

##### N-(4-(4,4-dicyano-3-(4-isopropylphenyl)butanoyl)phenyl)-4-methylbenzene-sulfonamide (**3k**)

White solid; (yield 0.48 g, 80%); mp: 175–176 °C; Rf 0.42 (30% EtOAc:Hexane); IR: 3450 (N-H str), 2966 (Ar C-H str), 1682 (C=O str), 1294 (asymm. S=O str), 1160 (symm. S=O str) cm^−1^; ^1^H NMR (400 MHz, DMSO-*d_6_*) (Appendix A) δ: 3.76 (s, 1H, NH), 7.85–7.83 (m, 2H, Ar-H), 7.73–7.71 (m, 2H, Ar-H), 7.36–7.30 (m, 4H, Ar-H), 7.23–7.20 (m, 4H, Ar-H), 5.14 (d, 1H, J = 5.80 Hz, H_1_), 3.95 (dd, 1H, J = 12.64 Hz, 6.48 Hz, 1H, H_2_), 3.68 (dd, 1H, J = 18.00 Hz, 3.24 Hz, 1H, H_3_) 3.50 (dd, 1H, J = 17.94 Hz, 11.92 Hz, H_4_), 2.87 (sept, 1H, J = 6.88 Hz, CH(CH_3_)_2_), 2.34 (s, 3H, Me), 1.20 (d, 6H, J = 6.88 Hz, CH(CH_3_)_2_); ^13^C NMR (100 MHz, DMSO-d6) (Appendix A) δ: 194.48, 148.2, 143.4, 142.78, 136.46, 134.83, 130.78, 129.57, 127.94, 126.65, 126.39, 117.62, 113.13, 112.87, 40.19, 38.94, 33.06, 29.03, 23.93, 20.98; CHN Analysis: Anal. calcd for C_28_H_27_N_3_O_3_S (485.60): C, 69.26; H, 5.60; N, 8.65%. Found: C1, 69.03; H, 5.39; N, 8.42%.

##### N-(4-(3-(2-chlorophenyl)-4,4-dicyanobutanoyl)phenyl)-4-methylbenzene-sulfonamide (**3l**)

White solid; (yield 0.50 g, 85%); mp: 182–184 °C; Rf 0.53 (30% EtOAc:Hexane); IR: 3458 (N-H str), 2966 (Ar C-H str), 1663 (C=O str), 1342 (asymm. S=O str), 1158 (symm. S=O str) cm^−1^; ^1^H NMR(400 MHz, DMSO-*d_6_*) (Appendix A) δ: 3.79 (s, 1H, NH), 7.85–7.83 (m, 2H, Ar-H), 7.73–7.71 (m, 2H, Ar-H), 7.61–7.59 (m, 1H, Ar-H), 7.49–7.47 (m, 1H, Ar-H) 7.36–7.30 (m, 2H, Ar-H), 7.22–7.20 (m, 2H, Ar-H), 5.24 (d, 1H, J = 5.56 Hz, H_1_), 4.54 (s,1H, H_2_), 3.77 (dd, 1H, J = 22.08 Hz, 8.48 Hz H_4_), 3.59 (dd, 1H, J = 18.08 Hz, 5.44 Hz, H_3_), 2.34 (s, 3H, Me); ^13^C NMR (100 MHz, DMSO-*d_6_*) (Appendix A) δ: 194.11, 143.46, 142.82, 136.43, 135.22, 133.97, 130.62, 129.62, 129.55, 129.5, 128.13, 127.47, 126.67, 117.61, 112.61, 40.19, 36.08, 27.82, 20.97; CHN Analysis: Anal. calcd for C_25_H_20_ClN_3_O_3_S (477.96): C, 62.82; H, 4.22;N, 8.79%. Found: C, 62.54; H, 3.98; N, 8.51%.

##### N-(4-(4,4-dicyano-3-p-tolylbutanoyl)phenyl)-4-methylbenzenesulfonamide (**3m**)

White solid; (yield 0.44 g, 77%); mp: 167–168 °C; Rf 0.48 (30% EtOAc:Hexane); IR: 3432 (N-H str), 2973 (Ar C-H str), 1682 (C=O str), 1334 (asymm. S=O str), 1162 (symm. S=O str) cm^−1^; ^1^H NMR(400 MHz, DMSO-*d_6_*) (Appendix A) δ: 3.75 (s, 1H, NH), 7.84–7.82 (m, 2H, Ar-H), 7.73–7.71 (m, 2H, Ar-H), 7.35–7.30 (m, 4H, Ar-H), 7.22–7.18 (m, 2H, Ar-H), 7.16–7.14 (m, 2H, Ar-H), 5.13 (d, 1H, J = 6.12, Hz, H_1_), 3.95 (dd, 1H, J = 6.16 Hz, 7.52 Hz, H_2_), 3.67 (dd, 1H, J = 17.95, Hz 7.96 Hz, H_4_), 3.49 (dd, 1H, J = 17.90 Hz, 6.04 Hz, H_3_), 2.34 (s, 3H, Me), 2.29 (s, 3H, Me); ^13^C NMR (100 MHz, DMSO-d6) (Appendix A) δ: 194.42, 143.38, 142.77, 137.42, 136.44, 134.43, 130.78, 129.53, 129.42, 129.02, 127.86, 126.64, 117.62, 113.08, 112.78, 40.2, 38.95, 29.02, 20.98, 20.63; CHN Analysis: Anal. calcd for C_26_H_23_N_3_O_3_S (457.54): C, 68.25; H, 5.07; N, 9.18%. Found: C, 68.02; H, 5.21; N, 8.98%.

##### N-(4-(3-(3-chlorophenyl)-4,4-dicyanobutanoyl)phenyl)-4-methylbenzene-sulfonamide (**3n**)

White solid; (yield 0.53 g, 90%); mp: 183–184 °C; Rf 0.52 (30% EtOAc:Hexane); IR: 3462 (N-H str), 1667 (C=O str), 1303 (asymm. S=O str), 1154 (symm. S=O str) cm^−1^; ^1^H NMR (400 MHz, DMSO-*d_6_*) (Appendix A) δ: 3.81 (s, 1H, NH), 7.86–7.84 (m, 2H, Ar-H), 7.74–7.72 (m, 2H, Ar-H), 7.54 (s, 1H, Ar-H), 7.42–7.30 (m, 5H, Ar-H), 7.23–7.21 (m, 2H, Ar-H), 5.26 (br s, 1H, H_1_), 4.03 (t, 1H, J = 6.68, Hz, H_2_), 3.73 (dd, 1H, J = 18.3 Hz, 3.0 Hz, H_4_), 3.53 (dd, 1H, J = 17.9 Hz, 11.96 Hz, H_3_), 2.35 (s, 3H, Me); ^13^C NMR (100 MHz, DMSO-*d_6_*) (Appendix A) δ: 194.21, 143.38, 142.9, 139.95, 136.46, 133.41, 130.63, 130.08, 129.53, 129.49, 128.16, 28.3, 126.65, 117.6, 112.76, 40.19, 38.94, 28.71, 20.98; CHN Analysis: Anal. calcd for C_25_H_20_ClN_3_O_3_S (477.96): C, 62.82; H, 4.22; N, 8.79%. Found: C, 62.54; H, 4.01; N, 8.51%.

#### 2.1.2. Biological Activity

##### Microfilariae Collection

The model animals *Meriones unguiculatus* (jirds) and *Mastomys natelansis* (mastomys) were used in this study to ensure patency of the filarial infection as per the guidelines of the Committee for the Purpose and Control of Experimental Animals. Microfilariae (Mf) were obtained by lavage of the peritoneal cavities of jirds with an intra-peritoneal filarial infection of 3 months or more duration. The collected Mf were washed with RPMI-1640 medium (ThermoFisher Scientific, Branchburg, NJ, USA) (containing 20 g/mL gentamycin, 30 g/mL penicillin, and 30 g/mL streptomycin), plated on sterile plastic Petri dishes, and incubated at 37 °C for 1 h to remove peritoneal exudate cells. The recovered Mf were then repeatedly washed using RPMI-1640 with antibiotics and used for in vitro experiments [14].

##### In Vitro Screening of Compounds for Antifilarial Activity

The efficacy of the compounds used to affect Mf viability in vitro was assessed by the extent of parasite motility. A stock solution of 2 mM concentration in DMSO was prepared for each chalcone derivative. Further dilutions were made in sterile isotonic sodium chloride solution to obtain a desired final concentration in the 0.5–500 μM range. The highest concentration of DMSO used with the compounds was <1%. Hence, 1% DMSO was used as a comparable vehicle control. Staurosporine (20 μM, standard apoptosis inducer) (Millipore Sigma, Darmstadt, Germany) was used as a positive control [15]. Approximately 300 Mf in 300 μL sodium chloride solution were introduced into each vial for each test drug (over a dose range of 1–30 μM) along with the vehicle control and incubated on a shaking incubator (Scigenics Biotech, India) at 37 °C for 30 min at 150 rpm. After incubation, the Mf were washed with RPMI-1640 medium and 30 Mf were plated in each well (each individual sample in triplicate) of sterile 24-well culture plates (Nunc, Denmark) containing 300 μL of RPMI-1640 medium. The plates were re-incubated at 37 °C for 48 h in a 5% CO_2_ incubator (pre-optimized conditions). Mf motility was assessed by microscopy (Nikon Diaphot, TMD inverted microscope, Tokyo, Japan). Each experiment was repeated thrice to check reproducibility. Percent inhibition in terms of loss of motility was determined as described earlier [13]. We selected IC_100_ for an effective antifilarial molecule to ensure the proposed complete apoptotic effect.

##### Determination of Lethal Dose of Chalcone Derivatives

The cytotoxicity of the chalcone derivative was evaluated by a trypan blue dye exclusion assay. Peripheral blood mononuclear cells (1 × 10^6^ cells/mL) derived from healthy human volunteers with informed consent were exposed to varying concentrations of compounds for 48 h followed by a 1 min incubation with trypan blue (0.2 mg/mL). The cells were observed under a Nikon light microscope (Tokyo, Japan), and the viable cell ratio was calculated by counting the stained and unstained cells separately. Viable cells do not take up trypan blue, while non-viable cells with porous membranes stain blue. The cytotoxicity of compounds was evaluated, and the 50% cytotoxic concentration (CC_50_) was determined [13].

##### Molecular Docking Studies

Since the three-dimensional structure of the *B. malayi* DHFR protein is not available in the Protein Data Bank (PDB), homology-dependent modeling was used to construct a three-dimensional protein structure using a homologous template protein (FZJ_A protein; GI:122920266), and structure validation was performed using ProSa-web, as previously described [16]. For molecular docking, a PDB file of the template protein was used. The molecular structure of the ligands was drawn using an online small-molecule topology generator (The GlycoBioChemPRODRG2 Server), after which molecular docking was performed using the AutoDock tool version 4.2. Consequently, the free energy levels of binding and inhibition constants were derived from the software-mediated analysis of the molecular docking.

##### DHFR Enzyme Assay

For the enzyme assay, all reagents were freshly prepared. DHFR activity was measured in the homogenate prepared from the untreated parasite. Mf extract was obtained by homogenizing the Mf in buffer A (0.5 mol/L Tris buffer, pH 7.5) containing a protease inhibitor cocktail (Sigma Aldrich, India) in a Remi type RQ127A homogenizer (Maharashtra, India). The homogenate was centrifuged at 5000× *g* for 20 min at 4 °C to remove debris and the resulting supernatant was further centrifuged at 30,000× *g* for 60 min at 4 °C. The supernatant containing the DHFR enzyme was collected and used for the enzyme assay.

Folic acid (FH_2_) stock solution (25 mg in 1.5 mL of 2-mercaptoethanol and 6.0 mL buffer A) was diluted in buffer B (0.05 mol/L Tris buffer, pH 7.5), yielding a final reaction solution of 0.34 g/L. The NADPH/DHFR reaction solution consisted of 0.4 mL NADPH stock solution (50 mg in 3 mL buffer A) and 0.8 mL Mf supernatant. FH_2_ reaction solution (130 µL) was added to each well of the 96-well flat-bottom plate. In the test well, compound **3g** (20 µL diluted in buffer B) was added at a final concentration of 38 µM in the reaction mixture. A control well was used that was devoid of any drug. The microplate was shaken on a plate shaker for 1 min; then NADPH/DHFR reaction solution (50 µL) was added to each well, and the microplate was shaken again on a plate shaker for 1 min. The absorbance of each well was read in a microplate reader (Agilent Synergy HTX Multi-Mode Reader, CA, USA) at 37 °C at 340 nm and 490 nm (reference) using a kinetic mode with a reading interval of 20 s for a duration of 18 min [17].

##### Folate Reversal Studies

Mf was pre-treated with 30 mM folic acid in RPMI-1640 medium for 1 h at 37 °C. Control Mf was incubated in RPMI-1640 medium only. After the incubation, the control and folic acid-pre-treated Mf were washed with RPMI-1640 medium and treated with **3g** at its IC_90_ concentration (34 μM) as described above. We used IC_90_ in this experiment because IC_100_ may permanently and irreversibly predispose Mf to apoptosis, which may not be suitable for observing the reversal of the antifilarial effect. The Mf were incubated for 48 h in 5% CO_2_ at 37 °C. Mf motility was assessed by microscopy.

##### MTT Assay

The control and **3g**-treated Mf (at IC_100_ concentration) were washed with 0.05 M phosphate-buffered saline (PBS, pH 7.2). The Mf were incubated in 30 µL of PBS containing 0.5 mg/mL MTT (Sigma Aldrich). After 2 h incubation, the Mf were washed by centrifugation, and DMSO was added to the Mf pellet to dissolve dark blue crystals of formazan. The mixture was transferred to a 96-well microtiter plate and read at 595 nm using DMSO as a blank [18]. 

##### Acridine Orange–Ethidium Bromide (AO/EB) Staining for Determination of Apoptosis

Dual staining with AO/EB was performed according to the standard protocol [13]. The dye mix consisted of 30 μg/mL AO and 30 μg/mL EB in phosphate-buffered saline. The Mf (negative control as well as Mf treated with **3g** or staurosporine for 48 h) were washed and re-suspended in 25 μL cold PBS, followed by the addition of 5 μL AO/EB dye mix. The stained Mf were viewed under a fluorescence microscope (Nikon E600 Fluorescence microscope, Tokyo, Japan) with the excitation filter set at 480/30 nm and the barrier filter at 535/40 nm. 

##### Cytochrome c ELISA

Vehicle- or compound-treated Mf were lysed with RIPA buffer (Himedia Laboratories Pvt Ltd., Maharashtra, India) for 1 h in the presence of protease inhibitors. The Mf lysates were centrifuged at 300× *g* for 3 min at 4 °C to remove cell debris, and the supernatants were centrifuged at 16,000× *g* for 20 min at 4 °C to pellet mitochondria and obtain a post-mitochondrial supernatant fraction.

The cytochrome c ELISA kit (Invitrogen, Maharashtra, India) was used to estimate cytochrome c protein content in the post-mitochondrial supernatant fraction as per the manufacturer’s instructions. Measurements were performed in duplicate, and the cytochrome c content was analyzed at 450 nm.

#### 2.1.3. Statistical Analysis

All experiments were performed in triplicate and the results are expressed as mean ± SD. Statistical significance was calculated using Student’s *t* test using SPSS version 16.0 (IBM, Armonk, NY, USA). The level of α error was limited to 5%.

## 3. Results and Discussion

### 3.1. Chemistry

The synthesis of Michael adducts was achieved by the reaction of sulfonamide chalcones **(1a-1n)** with compound **2** in moderate to high yields (70–96%) (Table 1). It has been observed that the substitution of the phenyl ring linked with β carbon of the double bond of reactant **1** has an influence on the rate of reaction. The electron-donating substituent posed some challenges, and lengthened the reaction time, while the electron-withdrawing substituent reduced the reaction time.

The structures of the newly synthesized Michael adducts were characterized by IR, ^1^H, ^1**3**^**C**, mass spectrometric and elemental analysis. The IR spectrum clearly indicates that N-H stretching at the absorption band was observed around 3200 cm^−1^, while a characteristic stretching frequency was observed at 2200–2250 cm^−1^ due to cyano groups. Characteristic symmetric and asymmetric stretching of the -SO_2_NH group were also observed around 1380 and 1350 cm^−1^, respectively. In the ^1^H spectrum, a broad singlet of N-H was observed around δ 3.8. There was a doublet around δ 6.0 for the H_1_ proton (Figure 1). The H_2_ proton resonated as a doublet or multiplets around δ 4.0. The H_3_ and H_4_ protons resonated as doublets of doublets around δ 3.9 and δ 3.6, respectively. The ^13^C NMR spectra showed the requisite number of distinct resonances in agreement with the designated structure. The ESI-MS of the compounds showed molecular ion peaks at their respective m/e values.

### 3.2. Biology

Chalcones, which are flavonoid metabolites, are expected to target the DHFR protein. Therefore, after minimizing the total interaction energies using a molecular dynamics program (obminimize tool, Open Babel), we studied the interaction of the Michael adducts (MA) of sulfonamide chalcones with *B. malayi* DHFR in silico. We found a measurably better interaction between these than what was observed with the basic structure of the chalcone compound (Table 2). Therefore, we further explored their effects on *B. malayi* in vitro. 

### 3.3. Antifilarial Activity and Cytotoxicity

The present work was designed to evaluate the antifilarial effects of chalcone derivatives and explore a possible antifolate action. The in vitro screening of 14 Michael adducts was performed, of which 4, namely, **3c**, **3g**, **3i**, and **3l**, showed pharmacological activity in terms of 100% loss of motility of all parasites in the culture (Table 3). This is in sharp contrast to DEC, which was shown to have no in vitro effect on the parasite. Other Michael adducts showed outcomes similar to that caused by the vehicle control, exhibiting no microfilaricidal activity up to 500 μM. The lowest IC_100_ value was observed with **3g**; therefore, we selected it for the mechanistic study. The IC_50_ value of **3g** was 23 μM.

The cytotoxicity of **3g** against human PBMCs was assessed by a trypan blue dye exclusion assay. The CC_50_ was 100 μM. These findings confirm the results of previous reports on various chalcone derivatives as potent antifilarial agents [13,19]. The sulfonamide chalcone derivatives used in this study are the products of the Michael addition reaction, reminiscent of the physiological xenobiotics disposal system that operates through glutathione-based Michael adducts [20]. The pharmacological significance of such agents as potential anticancer agents is also evident [21].

### 3.4. DHFR as a Target

The DHFR enzyme catalyzes the conversion of folic acid into dihydro-folic acid (FH_2_) and tetrahydro-folic acid (FH_4_), which are the feeders for the thymidine biosynthetic reaction in DNA synthesis. This makes DHFR a very lucrative drug target. As a proof of this principle, the approach to targeting this enzyme is exploited in the development of several antimicrobial and anticancer drugs, such as trimethoprim and methotrexate, respectively [22,23]. DHFR is present in numerous nematodes, including Dirofilariaimmitis, Litomosoides carinii, Dipetalonemaviteae, and Onchocerca volvulus [6]. Due to the paucity of research on *B. malayi*, the proteome database of this parasite lacks details on this protein. However, we found a genetic sequence of the *B. malayi dhfr* gene (EDP2873.1), and confirmed its actual expression as derived from the reported proteomic analysis [24]. In silico studies have proven DHFR to be a possible target for antifilarial drug development [16]. A similar bioinformatics-based approach was used to analyze the structure–activity relationship. As can be seen, all four compounds showed favorable docking in the nM range against *B. malayi* DHFR (Table 4). The presence of a p-tolyl group (4-Me-C6H4) in **3g** is responsible for the orientation of a sulfonyl group (O=S=O) of **3g** towards the Leu29 of BmDHFR to form a hydrogen bond (Figure 2). In contrast, the replacement of Me from a tolyl group with Cl (4-Cl-C6H4) in **3c** has been found to be responsible for the orientation of the O=S=O of **3c** in an opposite direction to Leu29. In the case of **3i** and **3l**, the presence of a 4-Me group on benzene-sulfonamide (which is absent in **3c** and **3g**) hindered the orientation of the O=S=O group towards Leu29. However, the presence of 2- or 4-chlorophenyl on the 4,4-dicyanobutanoylphenyl of **3c**, **3i**, and **3l** is responsible for these compounds’ antifilarial activity. A positive control epicatechin gallate (ECG) showed favorable binding against BmDHFR with a higher inhibition constant in the µM range. 

To validate the antifolate activity of **3g**, the DHFR activity in the Mf extract was determined spectrophotometrically using FH_2_ as a substrate in the presence of NADPH. As shown in Figure 3, a gradual decrease in absorbance over time was observed in a controlled reaction mixture (control 1) due to the consumption of chromogenic NADPH by active DHFR to convert FH_2_ to FH_4_. The addition of **3g** to the reaction mixture manifested a line parallel to the horizontal axis over time (a test), indicating the non-utilization of NADPH due to the **3g**-mediated inhibition of the DHFR activity. However, the initial absorbance level in this test is well below the corresponding level for control 1 (represented as a dashed line in Figure 3). To find out whether this is because of a possible interaction between **3g** and NADPH, only these two substrates (without Mf extract) were added in the buffer. The resulting absorbance was subtracted from the corresponding value of control 1 to obtain a second control line (control 2). This secondary control represents the available unbound chromogen (NADPH) in the presence of **3g**.

As shown in the figure, control 2 had an onset of absorbance equal to that of the test; however, it showed the expected gradual decline in absorbance over time. Therefore, it can be surmised that **3g** inhibits DHFR, as reflected by the trajectory of the higher absorbance by the unutilized NADPH in the test than in control 2 (after correction for the hypochromicity of NADPH due to the possible masking effect of chalcone on the actual absorbance of NADPH). This phenomenon can be attributed to a possible hindrance in absorbance by a purine base of NADPH due to its juxtaposition with the ring structure of chalcone, similar to the hypochromicity emerging due to the base stacking effect of hybridized DNA. This experiment was conducted per the previously described method [17], in which methotrexate was used as the standard DHFR inhibitor. Notably, methotrexate is an effective inhibitor of *B. malayi* DHFR [25,26]. Although the present experiment lacked a positive control (methotrexate), having considered similar procedures utilized in previous studies and in our work, we are confident about the accuracy of our experimental results, and believe that they will be quite reproducible. Based on this mechanistic insight, we will include this positive control in our future studies to develop a more potent and safer anti-filarial derivative.

To further confirm the antifilarial action of **3g** through DHFR inhibition, the possible reversal effect was studied using a DHFR substrate, folic acid. It was observed that folic acid could significantly reverse the antifilarial effect of **3g**. An almost 50% reduction in motility loss was observed in folic acid-pre-treated Mf, followed by **3g** treatment, compared with only **3g**-treated Mf (at IC_90_ dose), which showed the expected 90% loss of motility. This reversal in the presence of the substrate strongly suggests a mechanism of competitive inhibition carried out by the Michael adduct. As mentioned above, chalcones, which are structurally related to flavonoids, are expected to bear a resemblance to folate, a biological DHFR substrate. Therefore, **3g** may act as a competitive inhibitor of DHFR. 

### 3.5. Induction of Apoptosis

During the proliferative phase of the cell cycle, the demand for DNA synthesis increases. In this context, chalcone-induced DHFR inhibition may act as an apoptotic trigger. Therefore, our experimental evidence implies the necessity of detecting apoptosis as a consequence of DNA synthesis failure due to DHFR inhibition. For the detection of **3g**-induced apoptosis, the Mf were stained with EB/AO and observed under a fluorescence microscope (Figure 4). In EB/AO dual staining, AO permeates all cells, making the nuclei appear green, and EB is taken up by the cells only when the cytoplasmic membrane’s integrity is lost, staining the nuclei red. Therefore, viable cells have a normal green nucleus; early apoptotic cells have a bright green nucleus, with condensed or fragmented chromatin; late apoptotic cells show condensed and fragmented orange chromatin; cells directly killed by necrosis have a structurally normal orange nucleus. The Mf treated with staurosporine or **3g** showed orange–yellow fluorescence, indicating a loss of cell membrane integrity due to apoptotic damage. In contrast, the negative control (DMSO-treated Mf) was stained green, which implied an intact cell membrane.

Further, the MTT assay was performed to assess the **3g**-induced loss of cell viability. There was a 26.3% decrease in the formation of colored formazan in the **3g**-treated Mf (Figure 5). Both these results in congruence suggest the induction of the apoptotic process by this chalcone derivative.

A determination of cytoplasmic cytochrome c was carried out in order to yield confirmatory evidence of apoptosis. The Mf treated with **3g** or staurosporine showed markedly higher cytoplasmic cytochrome c release than that of the negative control (Figure 6). These results clearly substantiate the premise of apoptosis induction because of mitochondrial damage.

The results of the study confirm DHFR inhibition and the consequent apoptosis as the major mode of operation of chalcones against human lymphatic parasites. Based on the experimental findings, the role of the Michael adduct of sulphonamide chalcone as a potential chemotherapeutic agent against LF can be inferred. Future research goals include the further structural optimization of this molecule to increase its therapeutic index, as well as animal studies to establish it as a potent antiparasitic lead molecule.

## 4. Conclusions

Exploiting apoptotic impact through the structural analogy-based inhibition of DHFR was shown to be a successful method of antiparasitic drug development. The synthesis and characterization of sulfonamide chalcone-based Michael adducts, followed by the demonstration of their apoptotic rationale, reveals their therapeutic potential. 

## Figures and Tables

**Figure 1 biomedicines-11-00723-f001:**
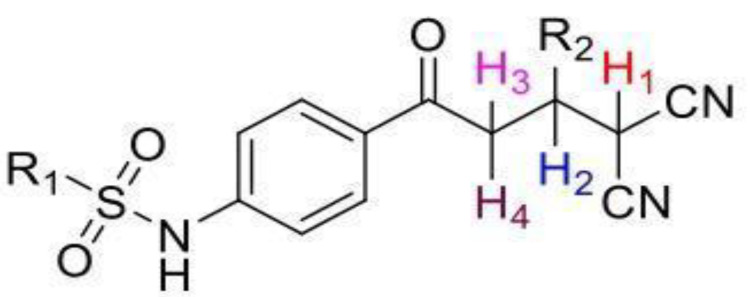
General representation of Michael adducts showing different hydrogens.

**Figure 2 biomedicines-11-00723-f002:**
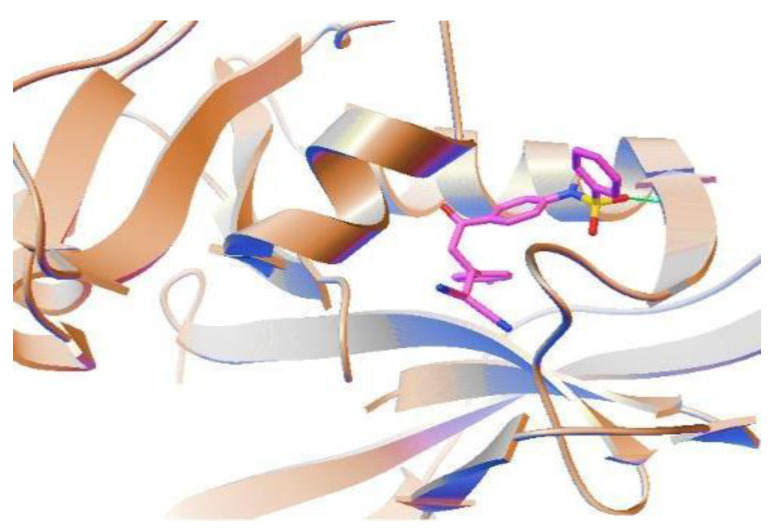
Molecular docking of **3g** with *Brugia malayi* DHFR protein. A p-tolyl group (4-Me-C6H4) directs a sulfonyl group (O=S=O) of **3g** towards Leu 29 of BmDHFR to form a hydrogen bond (green line).

**Figure 3 biomedicines-11-00723-f003:**
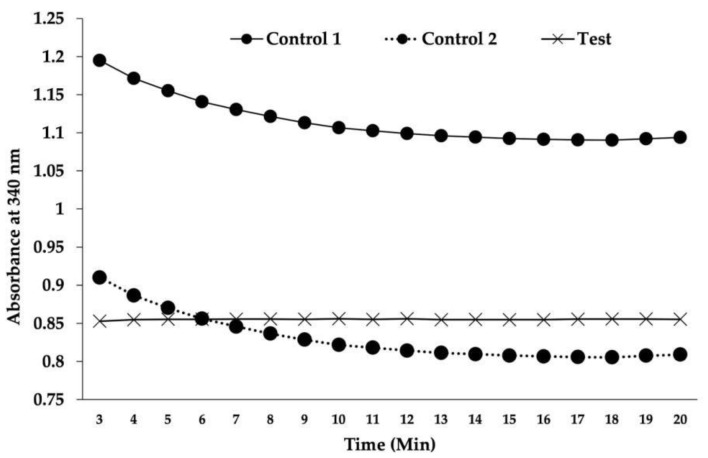
Change in absorbance of the reaction mixture at 340 nm over time. Data were obtained from 3 independent experiments set out to check for reproducibility. Control 1: Reaction mixture without **3g**. Control 2: Corrected control 1 line after deduction of absorbance obtained by drug-mediated hypochromicity of NADPH. **Test**: Reaction mixture with **3g** (38 μM).

**Figure 4 biomedicines-11-00723-f004:**
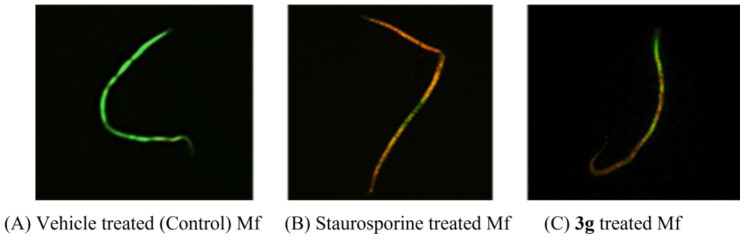
Acridine orange/ethidium bromide differential staining of Mf treated with (**A**) DMSO (a negative control), (**B**) staurosporine (20 μM, a positive control), or (**C**) **3g** (38 μM). Staurosporine or **3g**-treated Mf shows orange–yellow fluorescence, indicating apoptotic damage. In contrast, the DMSO-treated Mf was stained green, which indicates an intact cell membrane.

**Figure 5 biomedicines-11-00723-f005:**
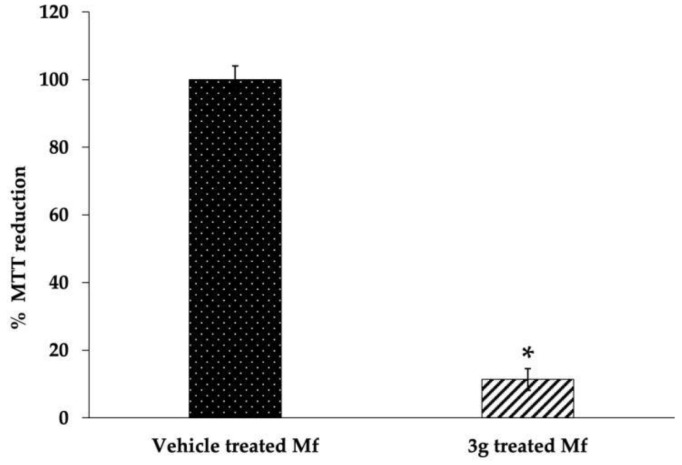
MTT reduction in **3g** (38 μM) treated Mf. Results are shown as mean ± SD. * *p* ≤ 0.005.

**Figure 6 biomedicines-11-00723-f006:**
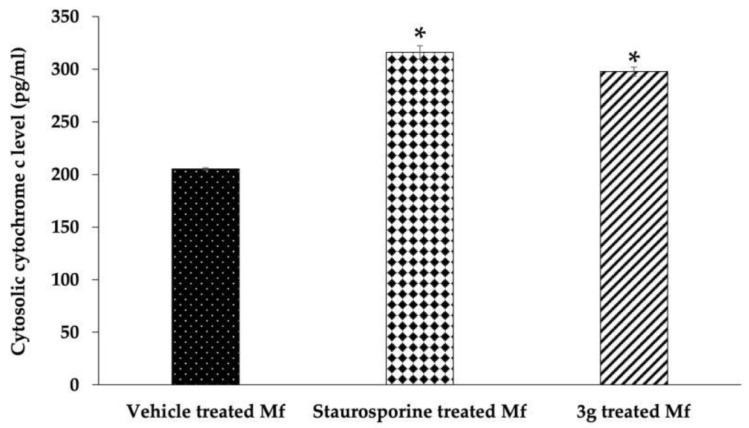
Cytosolic cytochrome c estimation in Mf treated with DMSO (a negative control), staurosporine (20 μM, a positive control), or **3g** (38 μM). Results are shown as mean ± SD. * *p* ≤ 0.05 compared to DMSO treated Mf.

**Table 1 biomedicines-11-00723-t001:** Synthesis of Michael adducts (**3a-3n**).

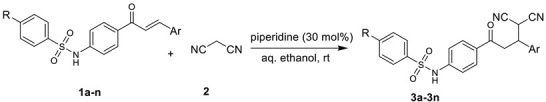
Entry	R	Ar	Product	Time (min)	Yield (%)	Mp (°C)
3a	H	-Ph	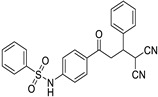	15	94	179–180
3b	H	-4-MeO-C_6_H_4_	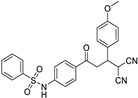	20	81	191–193
3c	H	-4-Cl-C_6_H_4_	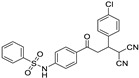	5	87	182–183
3d	H	-4-Br-C_6_H_4_	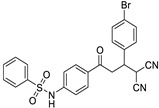	5	82	210–212
3e	H	-4-(MeO)_3_-C_6_H_2_	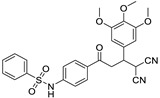	30	96	187–188
3f	H	-2-Cl-C_6_H_4_	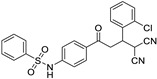	5	75	181–182
3g	H	-4-CH_3_-C_6_H_4_	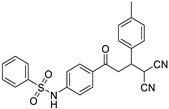	20	70	191–192
3h	-CH_3_	-4-MeO-C_6_H_4_	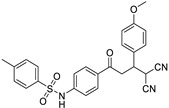	15	84	162–164
3i	-CH_3_	-4-Cl-C_6_H_4_	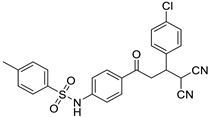	5	86	178–179
3j	-CH_3_	-4-Br-C_6_H_4_	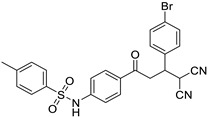	5	88	192–194
3k	-CH_3_	-3-iPr-C_6_H_4_	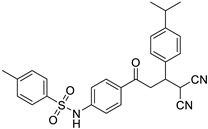	25	80	175–176
3l	-CH_3_	-2-Cl-C_6_H_4_	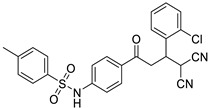	5	85	182–183
3m	-CH_3_	-4-CH_3_-C_6_H_4_	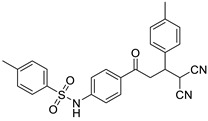	20	77	167–168
3n	-CH_3_	-3-Cl-C_6_H_4_	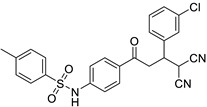	5	90	183–184

**Table 2 biomedicines-11-00723-t002:** In silico structure optimization of chalcone compound.

Compound	Chalcone	Michael Adduct (MA)
Structure	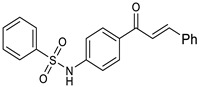	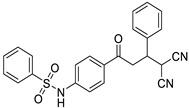
Free energy of binding (∆G_b_)	−6.93 kcal/Mol	−9.46 kcal/Mol
Inhibition constant (Ki)	8.31 µM	115.69 nM
Hydrogen bonding	Gly116:HN::O:Chalcone	Gly116:HN::OS:MA

**Table 3 biomedicines-11-00723-t003:** Antifilarial activity of Michael adducts of sulfonamide chalcone.

Compound	IC_100_
**3c**	114 ± 9 μM
**3g**	38 ± 1 μM
**3i**	132 ± 4 μM
**3l**	210 ± 0 μM

**Table 4 biomedicines-11-00723-t004:** Computationally derived free energy of binding and inhibition constant of Michael adducts of sulfonamide chalcones and ECG against BmDHFR target protein homology model.

Compound	∆Gb	Ki	Hydrogen Bonds
**3c**	−9.58 kcal/Mol	94.52 nM	No H-bond formed
**3g**	−9.54 kcal/Mol	101.83 nM	NH of Leu29 of BmDHFR: sulfonyl group (O=S=O) of **3g**
**3i**	−9.87 kcal/Mol	58.61 nM	No H-bond formed
**3l**	−9.41 kcal/Mol	125.65 nM	No H-bond formed
ECG	−7.82 kcal/Mol	1.84 µM	NH of Leu29 of BmDHFR: H of ECG

## Data Availability

Not applicable.

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
