# Peer review of "Michael Adduct of Sulfonamide Chalcone Targets Folate Metabolism in Brugia Malayi Parasite"

_biomedicines, 2023, doi:10.3390/biomedicines11030723_

Round 1
Reviewer 1 Report
Authors describe the synthesis and the biolgical evaluation of a small library of Michael adducts formed from chalcones. The experments and the results are convincing and the research seems to have appropriate novelty. However, there are still several questions and comments to be addressed.
1. In the abstract it is suggested to remove "3g" as ID it is too detailed.
2. In the experimental section some numbers should be in upper or lower index e.g. by the NMR and by the formula of the compounds.
3. NMR spectra should be povided as supporting information.
4. The general terms "online PRDRG server", "a molecular dinamics program" should be added more precisely.
5. Why did the authors apply IC100 or IC90 instead of the usual IC50?
6. Did the authors have the option to check the biological efficacy of the single enantiomers instead of the racemic mixture?
7. In Table 2 OMe or OCH3 should be unified to MeO
8. Chalcones are more flavonoid metabolites than derivatives.
9. In Table 4 the decimaly should be unified.
10. The concentration against human PBMC cells was not readable in the manuscript.
11. It is suggested to add a known positive control to Figure 3.
Author Response
- In the abstract it is suggested to remove "3g" as ID it is too detailed.
Reply: Per the Reviewer's suggestion, I have removed "3g" word throughout the abstract (lines 17, 22, 23, and 24).
- In the experimental section, some numbers should be in the upper or lower index e.g. by the NMR and by the formula of the compounds.
Reply: Necessary changes have been made.
- NMR spectra should be povided as supporting information.
Reply: NMR spectra has been provided in the supporting information.
- The general terms "online PRDRG server", "a molecular dinamics program" should be added more precisely.
Reply: I have now specified PRODRG as a "small-molecule topology generator (The GlycoBioChemPRODRG2 Server)" (line 241) and molecular dynamic program as "a molecular dynamics program (obminimize tool, Open Babel)" (line 315).
- Why did the authors apply IC100 or IC90 instead of the usual IC50?
Reply: The aim of the study is to explore the mechanism of anti-filarial (apoptotic) activity of 3g compound. Although we have checked for the usual IC50 dose (23 μM); however, to ensure the proposed apoptotic effect of the potential drug molecule, we have shown IC100 value, which is the minimum drug concentration at which 100% of the parasites used in the experiment were dead due to possible accomplishment of the complete apoptotic process.
In contrast, we used the IC90 value in the folate reversal study because we believed that IC100 may permanently and irreversibly predispose Mf to apoptosis, which might not be suitable for observing the reversal of antifilarial effect by folate.
- Did the authors have the option to check the biological efficacy of the single enantiomers instead of the racemic mixture?
Reply: Due to a lack in chiral HPLC facilities, we are unable to separate enantiomers and will consider this as a future scope for the study.
- In Table 2 OMe or OCH3 should be unified to MeO
Reply: Necessary changes have been made.
- Chalcones are more flavonoid metabolites than derivatives.
Reply: Thank you for suggesting a corrected term. We have made this change in the manuscript (lines 50 and 314).
- In Table 4 the decimaly should be unified.
Reply: We have now unified decimals.
- The concentration against human PBMC cells was not readable in the manuscript.
Reply: We are thankful for pointing out this mistake and regret the error. We have now corrected it (line 336).
- It is suggested to add a known positive control to Figure 3.
Reply: We do admit that adding a positive control might have helped to provide further resolution. However, due to the paucity of the available parasites from the infected model animal after lavage, it becomes difficult to manage enough parasites for all the experiments. Therefore, for certain experiments, such as DHFR assays, which need a very high number of parasites (150000 Mf), we could not manage to have positive controls.

Reviewer 2 Report
This is a very nice chemistry method development paper with the some biological activity and computational chemistry thrown in. The medicinal chemistry rationale for the structural variation is misrepresented, the in vitro data is incomplete, the compounds likely have substantial cytotoxic effects, and it is impossible to say if subsequent experiments supported the target ID conclusion (see below).
Page 2 Lind 67 – There is no synthesis info for compounds 1a-o. If they are new, then synthesis info needs to be provided. If they are not new, then a reference needs to be provided.
Page 2 Line 69 – Compound 1 should be changed to 1a-o.
Page 2 Line 73 - IUPAC name and number are missing for compound 3a.
Page 5 Line 226 – Cytotoxicity is expressed at CC50. Lethal dose (LD50) is for in vivo toxicity only.
Page 5 Line 235 – What protein was used as the starting point for the homology model? How does this protein compare to B. malayi DHFR?
Page 6 Scheme 1 – Why are there 15 starting compounds 1a-o but only 14 products 3a-n. Also compound numbering in scheme needs to be consistent (1a-o vs. 3a-3n).
Page 7 Table 2 – All of the chloro versions 3c, 3f, 3i and 3l have melting points of ~180 C except for 3n, which has a melting point of 220 C. Is this a mistake? If not, can the authors explain why this regioisomer would have a melting point difference of 40 degrees?
Page 9 Line 345 – The explanation for the rationalization for structural variation is not accurate. Medicinal chemists use an iterative approach to structural optimization where the biological results indicate the next compounds that should be synthesized. There is no data presented in this paper that indicates that malonitrile should be added to a chalcone Michael acceptor to create structural variation. It is clear that the authors made this series of compounds simply because they could (which is why the first part of the results section reads like an organic chemistry methods development paper) and then tested them for activity. This approach is fine. However, if this approach is taken, it is not appropriate to then state… “However, due to species-specific differential features of target protein, we optimized the chalcone structure for efficient targeting of parasitic DHFR.” And… “Thus, an in silico based molecular docking study provided insights into the ligand-target interaction and helped establish a framework for structural optimization. Accordingly, derivatives of MA were synthesized.” Instead, the authors should state that they made these compounds because they could but were pleasantly surprised to find that they also fit well in the proposed active site.
Page 10 Table 3 – If the data provided in the table is computationally derived, then this must be made clear in the title of the table. One should never leave any room for doubt about whether data is real or computationally derived.
Page 10 Line 373 – All Druglikeness rules are simply guidelines and should never be used as downselection criteria. Why not just say that 3g was favored simply because it was the only compound with an IC100 less than 100 micromolar?
Page 10 Table 4 – Where is the in vitro activity of the other compounds? What is the in vitro activity of a reference compound / positive control? Why is in vitro activity expressed as IC100 and not IC50 or IC90? IC100 is inherently problematic as it can imply a whole range of values… all the values that give 100% inhibition of growth. The authors should give some explanation for why IC100 was used. Also, the IC100 values presented show a very ‘flat’ SAR, indicating that no real structural optimization occurred here. Of course, it is impossible to say because only 4 of the possible 14 IC100 values are given. The authors must provide all IC100 values and objectively evaluate the resulting SAR.
Page 10 Table 5 – The title of the table should state that all of the information in the table is computationally derived. The name of the program used should also be given here either in the title or in a table legend. All of the PK information is essentially meaningless filler and should be removed as this kind of analysis should be reserved for virtual screening. The druglikeness analysis is also essentially meaningless, especially without any explanation of the Lipinksi, Veber, Egan and Muegge rules; it should also be removed.
Page 11 Line 386 – As mentioned previously, cytotoxicity is expressed as CC50. In vitro cytotox is not a complicated assay and should have been performed for all of the compounds; there was no reason to downselect prior to cytotox. There is a mistake in the unit on line 387, but I believe that the authors are saying that the CC50 of their lead compound was 100 micromolar. Unfortunately, their in vitro data was presented as IC100, so it is difficult to calculate an in vitro therapeutic index. However, given that the IC100 of compound 3g was 38 micromolar, the IVTI would be ~2.6. This is a terrible number as it indicates that some toxicity would likely occur around the IC100. More importantly, the fact that the CC50 value is lower than the IC100 for the other compounds in table 4 indicates that the observed inhibition of growth is simply a cytotoxic effect. Avoiding compounds that are cytotoxic is, after all, the whole reason for determining cytotoxicity.
Page 11 Table 6 – If the data provided in the table is computationally derived, then this must be made clear in the title of the table. One should never leave any room for doubt about whether data is real or computationally derived. Further, if the data results from the homology model then the authors must state that in the table title… ‘BmDHFR target protein homology model’.
Page 12 Figure 3 – What is the concentration of 3g used in this experiment? It is impossible to evaluate 3g’s activity here without knowing the concentration.
Page 12 Line 448 - It is not possible to say that 3g is a ‘potent’ inhibitor of DHFR based on the data provided in figure 3 and the IC100 value provided previously.
Page 12 Line 450 - Nothing about the data provided in figure 3 indicates anything about the potential ‘therapeutic safety’of 3g. Rather the cytotoxicity data provided earlier indicates just the opposite.
Page 13 Figure 4, Page 13 Figure 5, Page 14 Figure 6 – As in figure 3, concentrations of 3g used in these experiments needs to be provided.
Author Response
Comment 1: Page 2 Lind 67 – There is no synthesis info for compounds 1a-o. If they are new, then synthesis info needs to be provided. If they are not new, then a reference needs to be provided.
Reply: We have now added synthesis information with a reference (lines 67–69).
Comment 2: Page 2 Line 69 – Compound 1 should be changed to 1a-o.
Reply: We have changed compound 1 to 1a-n (line 72).
Comment 3: Page 2 Line 73 - IUPAC name and number are missing for compound 3a.
Reply: We are thankful for pointing out this mistake and regret for the error. We have now added it (line 77).
Comment 4: Page 5 Line 226 – Cytotoxicity is expressed at CC50. Lethal dose (LD50) is for in vivo toxicity only.
Reply: We thank the Reviewer for suggesting accurate terminology. We have now changed the word (lines 233 and 335).
Comment 5: Page 5 Line 235 – What protein was used as the starting point for the homology model? How does this protein compare to B. malayi DHFR?
Reply: For better clarity, we have now revised the sentence and cited a reference as follows: “Since the three-dimensional structure of the B. malayi DHFR protein is not available in the Protein Data Bank (PDB), homology-dependent modelling was used to construct a three-dimensional protein structure using a homologous template protein (FZJ_A protein; GI:122920266), and structure validation was performed using ProSa-web, as previously described [15].” (lines 237–240)
Comment 6: Page 6 Scheme 1 – Why are there 15 starting compounds 1a-o but only 14 products 3a-n. Also compound numbering in scheme needs to be consistent (1a-o vs. 3a-3n).
Reply: We regret for this typographic error; corrections have been made (lines 296–299).
Comment 7: Page 7 Table 2 – All of the chloro versions 3c, 3f, 3i and 3l have melting points of ~180 C except for 3n, which has a melting point of 220 C. Is this a mistake? If not, can the authors explain why this regioisomer would have a melting point difference of 40 degrees?
Reply: We thank the Reviewer for this careful observation. We checked all the database and needful correction have been made.
Comment 8: Page 9 Line 345 – The explanation for the rationalization for structural variation is not accurate. There is no data presented in this paper that indicates that malonitrile should be added to a chalcone Michael acceptor to create structural variation. It is clear that the authors made this series of compounds simply because they could (which is why the first part of the results section reads like an organic chemistry methods development paper) and then tested them for activity. This approach is fine. However, if this approach is taken, it is not appropriate to then state… “However, due to species-specific differential features of target protein, we optimized the chalcone structure for efficient targeting of parasitic DHFR.” And… “Thus, an in silico based molecular docking study provided insights into the ligand-target interaction and helped establish a framework for structural optimization. Accordingly, derivatives of MA were synthesized.” Instead, the authors should state that they made these compounds because they could but were pleasantly surprised to find that they also fit well in the proposed active site.
Reply: We do agree with the learned reviewer and admit that the compounds were actually synthesized in a series to be tested for their efficacy as antifilarial drug. However, considering the possible variations in the parasitic target protein, we tried to find more precise binding to the active site through bioinformatics tool and accordingly selected the compounds for actual experimental process rather than changing the molecular structure to optimize them. We have reframed the sentences to clarify the same in the revised version (lines 314–318).
Comment 9: Page 10 Table 3 – If the data provided in the table is computationally derived, then this must be made clear in the title of the table. One should never leave any room for doubt about whether data is real or computationally derived.
Reply: Thank you for the suggestion. We have revised the title now (now table 2, line 320).
Comment 10: Page 10 Line 373 – All Druglikeness rules are simply guidelines and should never be used as downselection criteria. Why not just say that 3g was favored simply because it was the only compound with an IC100 less than 100 micromolar?
Reply: We agree with the Reviewer. We have revised the sentence as follows: " The lowest IC100 value was observed for 3g; therefore, we selected it for mechanistic study." (line 328).
Comment 11: Page 10 Table 4 – Where is the in vitro activity of the other compounds? What is the in vitro activity of a reference compound / positive control? Why is in vitro activity expressed as IC100 and not IC50 or IC90? IC100 is inherently problematic as it can imply a whole range of values… all the values that give 100% inhibition of growth. The authors should give some explanation for why IC100 was used. Also, the IC100 values presented show a very ‘flat’ SAR, indicating that no real structural optimization occurred here. Of course, it is impossible to say because only 4 of the possible 14 IC100 values are given. The authors must provide all IC100 values and objectively evaluate the resulting SAR.
Reply: Thank you for pointing out this missed information. We have added information about other compounds in the text as follows: "Other Michael adducts showed outcomes similar to that of the vehicle control, exhibiting no microfilaricidal activity up to 500 μM." Moreover, IC100 value of a positive control is now mentioned in the text as follows "Staurosporine (20 μM, standard apoptosis inducer) was used as a positive control."
The aim of the study is to explore the mechanism of anti-filarial (apoptotic) activity of 3g compound. Although we have checked for the usual IC50 dose; however, to ensure the proposed apoptotic effect of the potential drug molecule, we have shown IC100 value, which is the minimum drug concentration at which 100% of the parasites used in the experiment were dead due to possible accomplishment of the complete apoptotic process.
In contrast, we used the IC90 value in the folate reversal study because we believed that IC100 may permanently and irreversibly predispose Mf to apoptosis, which might not be suitable for observing the reversal of antifilarial effect by folate.
Comment 12: Page 10 Table 5 – The title of the table should state that all of the information in the table is computationally derived. The name of the program used should also be given here either in the title or in a table legend. All of the PK information is essentially meaningless filler and should be removed as this kind of analysis should be reserved for virtual screening. The druglikeness analysis is also essentially meaningless, especially without any explanation of the Lipinksi, Veber, Egan and Muegge rules; it should also be removed.
Reply: We have deleted table 5 entirely, related to in silico druglikeness and pharmacokinetics study. Accordingly, the sections in the methods and results have also been deleted.
Comment 13: Page 11 Line 386 – As mentioned previously, cytotoxicity is expressed as CC50. In vitro cytotox is not a complicated assay and should have been performed for all of the compounds; there was no reason to downselect prior to cytotox. There is a mistake in the unit on line 387, but I believe that the authors are saying that the CC50 of their lead compound was 100 micromolar. Unfortunately, their in vitro data was presented as IC100, so it is difficult to calculate an in vitro therapeutic index. However, given that the IC100 of compound 3g was 38 micromolar, the IVTI would be ~2.6. This is a terrible number as it indicates that some toxicity would likely occur around the IC100. More importantly, the fact that the CC50 value is lower than the IC100 for the other compounds in table 4 indicates that the observed inhibition of growth is simply a cytotoxic effect. Avoiding compounds that are cytotoxic is, after all, the whole reason for determining cytotoxicity.
Reply: We are thankful to the reviewer’s comment. The corresponding values of the IC100, IC50, and CC50 of 3g compound used in this study were 38 μM, 23 μM, and 100 μM, respectively. According to the formula (CC50/IC50), IVTI is 4.3. Therefore, although the IC50 value is quite lower than that of the CC50 value, we admit that there is some chance of toxicity of this drug. By mechanistic proposition, the drug being a DHFR inhibitor the potential cytotoxic effect can be envisaged. Notwithstanding this fact, we humbly submit that as drug design/development follows an iterative approach to structural optimization based on biological results to achieve more specific binding with the target than the host counterpart, ensuring better therapeutic index, we will try to improve better compounds in future for which this present work will provide a strong foundation with mechanistic proof of concept (line 436).
Comment 14: Page 11 Table 6 – If the data provided in the table is computationally derived, then this must be made clear in the title of the table. One should never leave any room for doubt about whether data is real or computationally derived. Further, if the data results from the homology model then the authors must state that in the table title… ‘BmDHFR target protein homology model’.
Reply: We have now revised the title as "Computationally derived free energy of binding and inhibition constant of Michael adducts of sulfonamide chalcones and ECG against BmDHFR target protein homology model" (table 4, line 361).
Comment 15: Page 12 Figure 3 – What is the concentration of 3g used in this experiment? It is impossible to evaluate 3g’s activity here without knowing the concentration.
Reply: As described in the methods section, 3g compound (20 µL diluted in buffer B) was added at a final concentration of 38 µM in the reaction mixture. I have added this information in figure legends (line 383).
Comment 16: Page 12 Line 448 - It is not possible to say that 3g is a ‘potent’ inhibitor of DHFR based on the data provided in figure 3 and the IC100 value provided previously.
Reply: We have now revised a sentence as “Therefore, 3g may act as a competitive inhibitor of DHFR.” (line 397).
Comment 17: Page 12 Line 450 - Nothing about the data provided in figure 3 indicates anything about the potential ‘therapeutic safety’of 3g. Rather the cytotoxicity data provided earlier indicates just the opposite.
Reply: Thank you for pointing this out. We have deleted these sentences (line 397). We have already explained the cytotoxicity issue in the earlier response regarding this.
Comment 18: Page 13 Figure 4, Page 13 Figure 5, Page 14 Figure 6 – As in figure 3, concentrations of 3g used in these experiments need to be provided.
Reply: Per the Reviewer’s instructions, I have now added concentrations of staurosporine and 3g used in experiments.
Round 2
Reviewer 1 Report
Authors answered all the requested questions and comments. Although most of the answers are acceptable, there are two that in my view need further work.
It is accepted that for an appropriate number of experiments a high amount of parasites are necessary. I would recommend to repeat the experiment leading to figure 3 in order to have a positive control measurement.
Regarding IC90 and IC100 values, it is kindly asked from the authors to include their answer into the appropriate places of the manuscript and also add IC50 values at least in parentheses. It was difficult to follow the uploaded version of the manucript with the rack changes mode, so if it was already included, but not noticed, than this comment should be avoided.
Author Response
We are very much thankful for giving us an opportunity to modify our manuscript titled “Michael adduct of sulfonamide chalcone targets folate metabolism in Brugia malayi parasite” for a better presentation of our work with valuable inputs from the learned reviewers. We have tried our best to address the issues raised by them and accordingly modified the manuscript for a revised version (track changes on). The responses to all comments have been prepared and attached below.
Thank you for your consideration.

Reviewer 2 Report
The authors have done a really nice job of addressing the concerns that I brought up in my previous review.
Author Response

(The authors gave the same response as above.)

Round 3
Reviewer 1 Report
My opinion did not change, but as I see the authors also agree with the need of control experiments. It is a difficult situation that there is no money for that, for a referee it is not a great argument. From solidarity and human point it is acceptable. From a scientific point it is not. I trust the Editor to chose the more important point.
Author Response
Dear Reviewer,
We sincerely appreciate your input and support.
Thank you,
Kalyan Goswami